# Therapeutic Effect of Alpha Lipoic Acid in a Rat Preclinical Model of Preeclampsia: Focus on Maternal Signs, Fetal Growth and Placental Function

**DOI:** 10.3390/antiox13060730

**Published:** 2024-06-16

**Authors:** Gabriela Barrientos, Mariano L. Schuman, Maria S. Landa, Elizabeth Robello, Claudio Incardona, Melanie L. Conrad, Monica Galleano, Silvia I. García

**Affiliations:** 1Laboratorio de Medicina Experimental, Hospital Alemán, Ciudad Autónoma de Buenos Aires C1118AAT, Argentina; garcia.silvia@conicet.gov.ar; 2Consejo Nacional de Investigaciones Científicas y Técnicas (CONICET), Ciudad Autónoma de Buenos Aires C1118AAT, Argentina; 3Facultad de Medicina, Instituto de Investigaciones Médicas Alfredo Lanari, Universidad de Buenos Aires (UBA), Ciudad Autónoma de Buenos Aires C1053ABH, Argentina; marianoschuman@conicet.gov.ar (M.L.S.); mslanda@conicet.gov.ar (M.S.L.); 4Departamento de Cardiología Molecular, Instituto de Investigaciones Médicas (IDIM), UBA-CONICET, Ciudad Autónoma de Buenos Aires C1427ARN, Argentina; 5Facultad de Farmacia y Bioquímica, Cátedra de Fisicoquímica, Universidad de Buenos Aires (UBA), Ciudad Autónoma de Buenos Aires C1053ABH, Argentina; erobello@ffyb.uba.ar (E.R.); mgallean@ffyb.uba.ar (M.G.); 6Instituto de Bioquímica y Medicina Molecular-Dr. Alberto Boveris (IBIMOL), UBA-CONICET, Ciudad Autónoma de Buenos Aires C1113AAD, Argentina; 7Fundación GADOR, Ciudad Autónoma de Buenos Aires C1414CUI, Argentina; incardona@gador.com.ar; 8Institute of Microbiology, Infectious Diseases and Immunology, Charité Universitätsmedizin Berlin, Corporate Member of Freie Universität Berlin, Humboldt-Universität zu Berlin, Berlin Institute of Health, 12203 Berlin, Germany; conradml@gmail.com

**Keywords:** preeclampsia, placental dysfunction, animal model, antioxidant therapy, endothelial dysfunction, inflammation

## Abstract

Chronic hypertension is a major risk factor for preeclampsia (PE), associated with significant maternal and neonatal morbidity. We previously demonstrated that pregnant stroke-prone spontaneously hypertensive rats (SHRSP) display a spontaneous PE-like phenotype with distinct placental, fetal, and maternal features. Here, we hypothesized that supplementation with alpha lipoic acid (ALA), a potent antioxidant, during early pregnancy could ameliorate the PE phenotype in this model. To test this hypothesis, timed pregnancies were established using 10 to 12-week-old SHRSP females (*n* = 19–16/group), which were assigned to two treatment groups: ALA (injected intraperitoneally with 25 mg/kg body weight ALA on gestation day (GD1, GD8, and GD12) or control, receiving saline following the same protocol. Our analysis of maternal signs showed that ALA prevented the pregnancy-dependent maternal blood pressure rise (GD14 blood pressure control 169.3 ± 19.4 mmHg vs. 146.1 ± 13.4 mmHg, *p* = 0.0001) and ameliorated renal function, as noted by the increased creatinine clearance and improved glomerular histology in treated dams. Treatment also improved the fetal growth restriction (FGR) phenotype, leading to increased fetal weights (ALA 2.19 ± 0.5 g vs. control 1.98 ± 0.3 g, *p* = 0.0074) and decreased cephalization indexes, indicating a more symmetric fetal growth pattern. This was associated with improved placental efficiency, decreased oxidative stress marker expression on GD14, and serum soluble fms-like tyrosine kinase 1 (sFlt1) levels on GD20. In conclusion, ALA supplementation mitigated maternal signs and improved placental function and fetal growth in SHRSP pregnancies, emerging as a promising therapy in pregnancies at high risk for PE.

## 1. Introduction

Preeclampsia (PE) is a pregnancy-specific disorder that affects 2 to 8% of pregnancies worldwide and represents a leading cause of maternal and perinatal mortality [1]. PE usually manifests after the 20th week of pregnancy with hypertension, signs of maternal end-organ damage (i.e., kidney or liver dysfunction), and fetal growth restriction (FGR) [2]. There is growing awareness regarding the long-term consequences of PE, as it is associated with an increased risk of cardiometabolic disorders, renal disease, and other chronic conditions both in mothers and their offspring [3]. 

Chronic hypertension is a major risk factor for PE [4], and adverse maternal and perinatal outcomes are more prevalent when PE is superimposed on pre-existing hypertension than for de novo disease [5]. Management of this high-risk group is still challenging, as many women with chronic hypertension are recommended to modify their medication upon conception due to the potential teratogenic effects of some antihypertensive therapies [6]. Current treatment options include antihypertensive therapy to avoid progression to severe hypertension and low-dose aspirin for prevention of superimposed PE, but many women are non-responsive [7]. Furthermore, recent animal studies showed that standard antihypertensive therapies used in PE patients fail to improve maternal hemodynamics and cardiac compliance and, therefore, reduce maternal cardiovascular risk [8]. Therefore, the development of safe treatment alternatives that can help minimize the risk of adverse outcomes in hypertensive pregnancies is an acute need. 

While the origin of PE is still unclear, the placenta seems to play a central pathogenic role. Currently, PE is considered a multifactorial syndrome involving placental malperfusion with ensuing oxidative stress, an exaggerated inflammatory response, and angiogenic factor imbalance [9]. The release of antiangiogenic soluble fms-like tyrosine kinase 1 (sFlt1) by the ischemic placenta causes generalized endothelial dysfunction, intravascular inflammation, and activation of the coagulation system [10]. Targeting these mechanisms may be the key to developing an effective pharmacological treatment for pregnancies affected by PE. In particular, antioxidant treatment (i.e., with vitamins E/C, melatonin, or other supplements) has long been proposed as an option for PE. However, the results of the many trials conducted so far have been disappointing [11]. Over 15 trials in the past two decades have evaluated the use of antioxidant supplements in hypertensive disorders of pregnancy (as reviewed in [11]) but showed no conclusive evidence of their beneficial effect. In this context, the ‘master antioxidant’ alpha lipoic acid (ALA) has been hypothesized as a potential candidate for the treatment of PE due to its diversity of effects, including among others the ability to scavenge free radicals and regenerate endogenous antioxidants [12], promote the expression of antioxidant enzymes [13], mitigate endoplasmic reticulum stress [14], inhibit inflammatory signaling cascades [15,16] and enhance nitric oxide availability [17]. However, to date, this hypothesis lacks formal testing in preclinical or clinical settings. 

Preclinical animal models represent an essential prerequisite for the validation of novel therapeutic interventions. Our previous studies have described the stroke-prone spontaneously hypertensive rat (SHRSP) as a preclinical model of PE with distinct placental, maternal, and fetal features [18,19,20]. Pregnant SHRSP spontaneously display increased blood pressure, signs of renal compromise including decreased urine output, and eventual proteinuria and asymmetric FGR [18]. These features develop secondarily to impaired placentation with defective trophoblast invasion [18], increased production of sFlt1, and activation of placental oxidative stress pathways [19,20]. Considering these findings, we hypothesized that ALA supplementation during the placentation period would ameliorate PE-like features in pregnant SHRSP [21]. To test this hypothesis, we evaluated the effects of ALA treatment on maternal blood pressure and renal function parameters, fetal outcomes, placental morphology, and placental levels of oxidative stress, inflammation, and angiogenic markers.

## 2. Materials and Methods

An expanded Materials and Methods section is available as Appendix A.

### 2.1. Animals and Experimental Design

Timed pregnancies were established using SHRSP females (10–12 weeks old, 200–250 g body weight *n* = 19–16 animals/group) as described previously [18]. Matings were established so as to achieve an *n* = 8–10 animals/treatment group for each time point analyzed. The number of animals was based on previously published studies using the same model [18,19]. Females were randomly paired to littermate males (at a 1:1 ratio) using a computer-based random order generator. Upon vaginal plug detection (GD1), dams were assigned to two groups: ALA treatment (injected intraperitoneally with 25 mg/kg ALA on GD1, GD8, and GD12) and controls, receiving saline under the same protocol. Allocation to the treatment groups was performed using a minimization strategy so that there was a balanced distribution of animals to the treatment groups throughout the study. The timing of ALA administration was set empirically to encompass the placentation period (which is completed by GD12) while minimizing animal distress during the peri-implantation period due to manipulation and injection. The treatment dose was set in agreement with previous reports on the administration of ALA to pregnant rats [22]. Maternal blood pressure profiles throughout pregnancy were recorded using a noninvasive volume pressure recording device, as reported previously [18]. Briefly, blood pressure records were taken at baseline (pre-mating), GD1, GD7, GD14 (onset of disease), and GD18 (late gestation, fully established maternal syndrome). Records were made in the morning, with five cuff inflations as acclimation followed by 20 measurement cycles. Animals were euthanized on GD14 or GD20 for collection of fetoplacental specimens and kidneys used for morphological and molecular analyses. Maternal weight and the specific weight gain (normalized to litter size) during pregnancy were determined on GD20. Further details of the experimental protocol, including raw data for blood pressure profiles and fetal growth parameters, are available in the Appendix A.

### 2.2. Sex as a Biological Variable

Maternal parameters (BP, weight gain, renal function, and serum biomarkers) were determined in females because the disease modeled is only relevant in females. For fetal and placental determinations, sex was not considered as a biological variable.

### 2.3. Fetal Phenotype Evaluation

Fetal development was assessed morphometrically on GD20. Briefly, following the collection of amniotic fluid, the amnion was dissected to expose the fetuses and placentas, which were fixed in phosphate-buffered 10% formaldehyde (pH 7.2). Placental and fetal weights were recorded from the fixed specimens, and fetal head diameter was measured using a Vernier caliper. The cephalization index (head circumference to body weight ratio) was calculated as an indicator of asymmetric fetal growth, as described in our previous publication [18].

### 2.4. Histology and Immunohistochemistry

Morphometrical evaluation of placental and kidney samples was performed as described in our previous publication [18], using QuPath v0.4.4 (https://qupath.github.io/, accessed on 23 April 2024) [23] and Angiotool (https://ccrod.cancer.gov/confluence/display/ROB2/Home, accessed on 23 April 2024) [24] for quantitative image analysis. 

### 2.5. Western Blots

Expression of NLRP3, SOD2, and NOX 1 and 2 subunits (NOX1, NOXO1, p47phox, gp91phox) was evaluated in placental extracts following a previously published western blot protocol [19]. Images of the full unedited blots are included in the Appendix A.

### 2.6. Placental Gene Expression Analysis

Quantitative real-time reverse transcriptase polymerase chain reaction (RT-PCR) was used to evaluate the mRNA levels of caspase 1 (*Casp1*), interleukin (IL) 1β (*Il1b*), IL18 (*Il18*), Flt1 (*Flt1*), vascular endothelial growth factor (*Vegfa*), and hypoxia-inducible factor 1 alpha (*Hif1a*) as published [25]. Detailed protocols and primer sequences are included in the Appendix A.

### 2.7. Statistical Analysis

Results are expressed as mean ±SD or median (minimum–maximum), with n representing the number of individual experiments, pregnant dams, or fetuses. Outliers were identified using the ROUT method (Q = 5%) [26], which allows simultaneous detection of multiple outliers based on robust nonlinear regression and the false discovery rate. Data were checked for normality using the Shapiro–Wilk test. Student’s *t*-test or Mann–Whitney U-tests were used to analyze normally distributed and skewed data, respectively. Graphing and calculations were performed using the GraphPad Prism 8.0 software (https://www.graphpad.com/, accessed on 23 April 2024). A *p*-value < 0.05 was considered statistically significant. 

## 3. Results

### 3.1. Treatment with ALA Prevents Maternal Blood Pressure Elevation and Mitigates Renal Damage in SHRSP Pregnancies

To investigate its therapeutic potential in PE, we designed a protocol for ALA supplementation in the SHRSP model (Figure 1A) and evaluated the effects of treatment on maternal signs, fetal growth, and placental function. Compared to the control treatment, ALA prevented the pregnancy-dependent maternal blood pressure elevation in the SHRSP model, as shown by the significantly decreased systolic blood pressure observed on GD14 (169.3 ± 4.3 mmHg vs. 146.1 ± 3.4 mmHg, *p* = 0.0001, Figure 1B and Appendix A). Treated dams also showed a significant increase in the creatinine clearance (Cr Cl) on GD20 (control 1.36 ± 0.17 mL/min vs. ALA 1.62 ± 0.15 mL/min, *p* = 0.037, Figure 1C), but no differences were observed in the albumin-to-creatinine ratio (ACR, 0.51 ± 0.07 vs. 0.54 ± 0.08, *p* = 0.4705) or urine output compared to controls (Figure 1C and Appendix A). Histological evaluation of the kidneys on GD20 revealed a marked decrease in mesangial extracellular matrix accumulation in treated dams (median PAS-positive area control 53.81% vs. ALA 30.29%, *p* < 0.0001, Figure 1D, left panel), but no significant differences in the glomerular expression of podocin (*p* = 0.1213, Figure 1D, right panels). The treatment produced no significant effect on maternal weight or the specific weight gain during pregnancy (*p* = 0.8598 and *p* = 0.4412, respectively, Appendix A).

### 3.2. ALA Supplementation Rescues the Asymmetric FGR Phenotype in the SHRSP Model

Since SHRSP pregnancies are characterized by asymmetric FGR [18,27], we next assessed biometric parameters on GD20 to determine the effect of ALA on the fetal phenotype. Figure 2A shows representative examples of fetuses collected from ALA-treated dams (left panel), which displayed significantly increased body weights compared to the controls (2.19 ± 0.28 g vs. 1.98 ± 0.48 g, *p* = 0.0074, Figure 2A). Furthermore, in an analysis performed to verify the reproducibility of our previous observations [18], we observed that ALA treatment was able to restore fetal weight in the SHRSP model, showing no significant differences compared to the normotensive control strain (Appendix A). The cephalization index, which, when increased, indicates a ‘brain-sparing’ effect consistent with asymmetric FGR [27], was significantly decreased upon ALA treatment (control 1.63 ± 0.19 cm/g vs. ALA 1.53 ± 0.24 cm/g, *p* = 0.0181, Figure 2B and Appendix A). The improved fetal phenotypes observed in ALA-treated animals were associated with a significant increase in placental efficiency (measured as the fetal to placental weight ratio, control 4.92 ± 0.13 vs. ALA 5.49 ± 1.17, *p* = 0.0453, Figure 2C) and placental weight (control 0.39 ± 0.05 g vs. ALA 0.41 ± 0.05 g, *p* = 0.0236, Appendix A).

### 3.3. Treatment with ALA Attenuates Placental Oxidative Stress and NLRP3 Inflammasome Priming

Having observed an increased placental efficiency, we next hypothesized that the beneficial effect of ALA supplementation would be related to an improved placental environment in SHRSP pregnancies. Based on our previous findings showing an enhanced activation of oxidative stress pathways [19], we first evaluated the placental expression of NADPH oxidase (NOX) isoforms on GD14. ALA treatment led to a significant decrease in the expression of the cytosolic subunit NOXO1 (relative expression 1.00 ± 0.28 vs. 0.55 ± 0.08, *p* = 0.0286, Figure 3A) versus controls. However, we observed no differences in the expression levels of p47phox (relative expression 0.89 ± 0.29 vs. 0.93 ± 0.22, *p* = 0.9762, Figure 3A), NOX1, gp91phox (*p* = 0.0952 and *p* = 0.2857, respectively, Appendix A) or the antioxidant SOD2 (relative expression 1.00 ± 0.19 vs. 0.85 ± 0.09, *p* = 0.3429, Figure 3B). 

Since NOX-derived ROS are known to promote NLRP3 inflammasome priming [28], we evaluated the expression of components of this pathway in the GD20 placenta. ALA treatment significantly decreased the protein expression of NLRP3 in SHRSP placentas (relative expression 1.00 ± 0.14 vs. 0.42 ± 0.27, *p* = 0.0159, Figure 3C) as well as decreased mRNA levels of caspase 1, IL1β and IL18 (*p* = 0.0286, *p* = 0.0286 and *p* = 0.0286, respectively, Figure 3C). Further analysis of amniotic fluid collected on GD20 revealed that ALA did not affect levels of thiobarbituric acid reactive substances (TBARS, *p* = 0.5605, Appendix A) observed in SHRSP pregnancies but induced a marked decrease in IL18 protein levels (control 77.2 ± 26.8 pg/mg protein vs. ALA 32.0 ± 11.4 pg/mg protein, *p* = 0.0068, Appendix A).

### 3.4. ALA Improves Placental Bed Vascular Remodeling and Labyrinth Vascularization in SHRSP Pregnancies 

Next, we assessed the effect of ALA on placental dysfunction in the SHRSP model, which is characterized by defective spiral artery remodeling and impaired trophoblast differentiation associated with altered angiogenic signaling [18]. Histological evaluation of GD14 placentas showed that ALA treatment improved the physiological remodeling of spiral arteries, as shown by the significant loss of arterial wall ACTA2 expression in the mesometrial triangle of treated placentas (median expression control 63.12% vs. ALA 31.7%, *p* ˂ 0.0001, Figure 4A left panels). Differentiation of the placental junctional zone was also restored in ALA-treated dams, denoted by the increased accumulation of PAS-positive glycogen trophoblasts compared to controls (median PAS-positive area control 29.29% vs. ALA 35.20%, *p* = 0.015, Figure 4A right panels). On GD20, treated dams displayed an increased vascular density in the placental labyrinth (median Isolectin B4(IB4)-positive staining control 26.04% vs. ALA 27.96%, *p* = 0.0284, Figure 4B). ALA decreased the placental mRNA expression of Flt1 and HIF1a (*p* = 0.0286 and *p* = 0.0079, respectively) but had no significant effect on levels of VEGF (*p* = 0.2286, Figure 4C). Compared to controls, serum levels of FLT1 on GD20 were significantly decreased in ALA-treated dams (control 120.0 ± 8.6 pg/mL vs. ALA 102.6 ± 11.3 pg/mL, *p* = 0.0286, Figure 4D).

## 4. Discussion

In this study, we demonstrated the protective effect of ALA supplementation in a rat model of hypertensive pregnancy with spontaneous progression to PE. In pregnant SHRSP, treatment with ALA during early pregnancy improved maternal blood pressure control and restored fetal growth associated with attenuated placental oxidative stress, inflammation, and antiangiogenic status. Given its beneficial effect on both maternal and fetal outcomes, ALA emerges as a promising therapeutic agent in pregnancies at high risk for PE. 

The beneficial effects of ALA on cardiometabolic health, as well as its mechanisms of action, have been extensively studied [29,30]. Furthermore, evidence on the clinical use of ALA in pregnancy [31,32,33], including pharmacovigilance upon oral supplementation [34], has accumulated over the past decade, supporting its potential as a safe therapeutic intervention in PE. By showing an improved blood pressure regulation upon ALA treatment in pregnant rats with chronic hypertension, our preclinical findings are potentially translatable to a high-risk patient population that still imposes a challenge in terms of management. Current guidelines recommend antihypertensive therapy to limit the risk of progression to severe hypertension and low-dose aspirin and calcium for prevention of superimposed PE [35,36], but none of these strategies have proven definite in avoiding adverse maternal and perinatal outcomes. Antioxidant supplementation, in particular, has long been proposed as a therapeutic option for PE, but the many large-scale randomized clinical trials run so far have failed to demonstrate positive effects [11]. Our results in a preclinical model suggest that ALA may be successful where other antioxidants have failed, and this may be due to its diversity of molecular targets and inhibitory effects on cellular stress responses, inflammation, and endothelial dysfunction [12]. Regarding its effects on blood pressure, hypotensive effects upon ALA supplementation have been reported in animal studies [37,38] but are attained upon chronic treatment or single doses much higher than the one used in our study. Hence, the observed blood pressure-lowering response in SHRSP pregnancies may be secondary to the improved placental environment rather than a direct effect of ALA on the maternal vasculature. Collectively, these evidences encourage further investigation into the effect of ALA supplementation either alone or in combination with the currently recommended treatments in pregnant women with chronic hypertension. 

A major finding in our study was the positive effect of ALA on placental insufficiency, which was noticed as early as GD14 through the improvement of spiral artery remodeling and differentiation of glycogen trophoblasts. In the SHRSP model, defective arterial remodeling [18] would result in intermittent high-pressure flow reaching the placenta, contributing to ischemia-reperfusion injury and ensuing oxidative stress. Sustained ROS production could then compromise cell organizational processes related to trophoblast differentiation, resulting in the marked structural alterations observed both in the SHRSP and in the ischemic placenta of the reduced uteroplacental perfusion pressure (RUPP) PE model [39,40]. Hence, it is likely that the observed positive effect of ALA on placental insufficiency in our model is related to the multiple mechanisms by which it can prevent ROS accumulation [29], which have encouraged its therapeutic application against ischemia-reperfusion injury in both clinical and basic settings [41,42,43]. 

Further considering oxidative damage, glycogen trophoblasts seem to be particularly vulnerable [40] and display an impaired migratory potential in the ischemic RUPP placenta [39]. Glycogen trophoblast injury has major overall consequences in placental development and function since, at later stages, these cells invade the maternal compartment to further support arterial remodeling [44] and influence the differentiation of other placental lineages by producing retinoic acid [45]. Given that these cells are considered analogous to the human extravillous trophoblast [46], an intriguing question arising from our study is whether ALA can exert a direct influence on human trophoblast differentiation, migration, and invasive properties. Other than a study reporting an inhibitory effect of ALA on hypoxia-induced apoptosis of a villous trophoblast-derived cell line [47], evidence of its in vitro effects on human trophoblasts is still lacking. 

During early pregnancy, placental ROS production occurs mainly through NOX activation and contributes to healthy trophoblast function by modulating redox-sensitive MAPK signaling pathways [48]. However, sustained activation of NOX plays a central pathogenic role in PE as it drives the placental secretion of sFlt1 [49] and may activate the NLRP3 inflammasome [28,50], promoting maternal systemic endothelial dysfunction and inflammation. Thus, the decreased placental expression of cytosolic NOXO1 (the organizer protein required for catalytic activation [51]) we observed upon ALA treatment suggests that inhibition of NOX-mediated ROS generation may be core to the protective effect of ALA in SHRSP pregnancies. Besides suppressing NOX activation, the effect of ALA on sFlt1 secretion could also be due to its potent ability to induce heme-oxygenase 1 activity [52], as this pathway has been shown to negatively regulate the release of sFlt1 and soluble endoglin from villous explants [53]. Indeed, considering the treatment timeframe in our study, it is likely that the antioxidant effects of ALA improved the physiological adaptation of the SHRSP placenta to the oxidative burst associated with the onset of maternal blood flow to the hemochorial interface, which in the rat occurs around GD12 [54]. This is highlighted by the finding that the inhibitory effects we observed on sFlt1, IL1β, and IL18 persisted on GD20 beyond the temporal window of ALA administration. 

Besides mitigating maternal signs, it is desirable that a PE therapeutic agent would prevent adverse fetal outcomes associated with FGR. Here, we demonstrated the ability of ALA to rescue asymmetric FGR in the SHRSP model, which is consistent with previous animal studies reporting similar effects in models of diabetic [55] and toxicant-induced intrauterine growth restriction [56]. Asymmetric FGR, as observed in the SHRSP [18,27], results from impaired nutrient transfer due to placental dysfunction and thus shows a strong association with PE [57]. The protective effect of ALA on fetal growth likely stems from improved placental efficiency for metabolic exchange, as suggested by the increased fetal-to-placental weight ratios and enhanced labyrinth vascularization observed upon ALA treatment. However, an interesting finding of our study is that ALA treatment was associated with decreased levels of IL18 in the amniotic fluid, suggesting that the anti-inflammatory effects of ALA can also reach the fetal compartment via maternal supplementation. Since growth factors and signaling molecules in the amniotic fluid can be absorbed by the fetus and potentially influence its development [58], our study opens an exciting avenue of research into the developmental programming effects induced by maternal ALA supplementation. Recent studies, for instance, have demonstrated that maternal ALA treatment can attenuate the metabolic alterations associated with the consumption of high-fructose diets in rat offspring [59], supporting a role for ALA in programming cardiometabolic health in utero. 

Translational research remains a challenge for both bench scientists and physicians, particularly in obstetrics, due to concerns related to safety and teratogenicity. While further research involving larger cohorts should be encouraged, a retrospective study of 610 pregnancies reported no adverse effects in the mother or newborn upon daily oral administration of 600 mg ALA for at least seven weeks [34], opening a reassuring scenario regarding its therapeutic use during pregnancy. Based on an allometric scaling approach for interspecies dose conversion [60], 600 mg represents a dose higher than the one administered in our study, adding to the clinical relevance of our findings. However, there are limitations inherent to the animal model that need to be taken into account when extrapolating our results. The main consideration is that unlike in PE subjects, blood pressure in hypertensive rat models decreases towards delivery, as reported in the SHRSP and other strains [61]. Furthermore, we acknowledge that no single animal model of PE can fully replicate the human syndrome due to disease heterogeneity, diversity of molecular signatures, clinical manifestations, and outcomes [62]. However, the pregnant SHRSP phenotype not only fulfills the current definition of PE, combining hypertension with end-organ damage or FGR [2], but also presents several important traits associated with PE pathogenesis, including defective spiral remodeling, placental dysfunction, and angiogenic imbalance. Given its effects in mitigating maternal signs and restoring placental function in this model, we propose that ALA supplementation represents a promising intervention to improve the outcomes of pregnancies at high risk for PE, for which there is currently no definite therapeutic option. 

## 5. Conclusions

In conclusion, ALA supplementation mitigated maternal signs and rescued the placental dysfunction and FGR associated with the development of PE in our preclinical model of hypertensive pregnancy. Our preclinical observations suggest that ALA could be successful where other antioxidants have failed and encourage further investigation into the effect of ALA supplementation on pregnant women with chronic hypertension, either alone or in combination with the currently recommended treatments. Further studies should address the therapeutic use of ALA in other preclinical models of PE or isolated FGR, as well as examine the effects on human trophoblast biology in vitro. This knowledge, together with accumulating clinical evidence on the safety of ALA administration in pregnancy, set the ground for further studies aimed at repositioning ALA as a preventive therapy for pregnancies at high risk for PE.

## Figures and Tables

**Figure 1 antioxidants-13-00730-f001:**
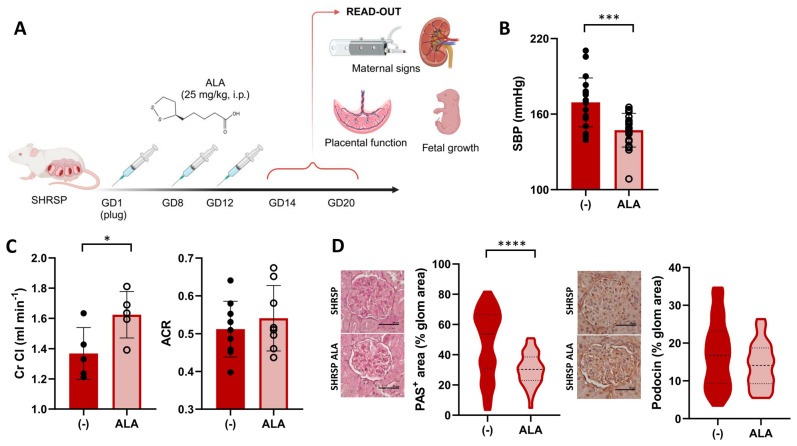
ALA improves maternal PE signs in SHRSP pregnancies. (**A**) Scheme of the experimental protocol for ALA supplementation. Panel created with Biorender (**B**) Maternal systolic BP (SBP) in ALA-treated SHRSP, measured on GD14. Data are individual measurements and mean ± SD, *n* = 20–21/group, *** *p* ˂ 0.001 analyzed by *t*-test. (**C**) Creatinine clearance (Cr Cl, left graph) and urine albumin-to-creatinine ratio (ACR, right graph) assessed on GD20. Data are individual measurements and mean ± SD, *n* = 5–9/group, * *p* ˂ 0.05 analyzed by *t*-test. (**D**) Histological assessment of renal damage in ALA-treated dams. Mesangial matrix expansion (PAS-positive stain, left graph) was decreased upon ALA treatment. Glomerular expression of podocin (right graph), assessed by immunohistochemistry, was not modified by the treatment. Scale bar: 50 µm. Data are medians (dashed line) and minimum–maximum, *n* = 5–6 samples/group analyzing 40–100 glomeruli using QuPath. **** *p* ˂ 0.0001 analyzed by Mann–Whitney U-test.

**Figure 2 antioxidants-13-00730-f002:**
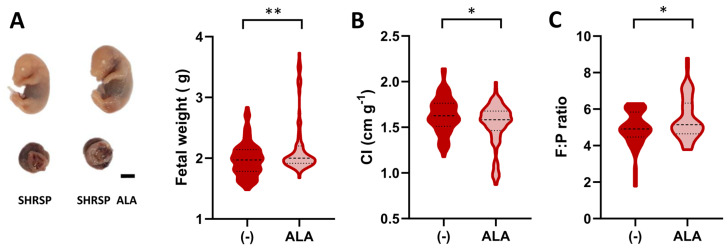
Effects of ALA supplementation on fetal phenotype. (**A**) Representative examples of formalin-fixed fetal and placental specimens collected on GD20. Scale bar: 5 mm. Fetal weight (violin plot) was significantly increased in the ALA-treated group. (**B**) Fetal cephalization indexes (CI, head circumference/body weight) measured on GD20. (**C**) Fetal to placental (F:P) weight ratio was measured in both treatment groups. Data in all panels are medians (dashed line) and minimum–maximum. (**A**,**B**): *n* = 88–85 specimens. (**C**): *n* = 53–54 specimens. ** *p* ˂ 0.01, * *p* ˂ 0.05 analyzed by Mann–Whitney U-test.

**Figure 3 antioxidants-13-00730-f003:**
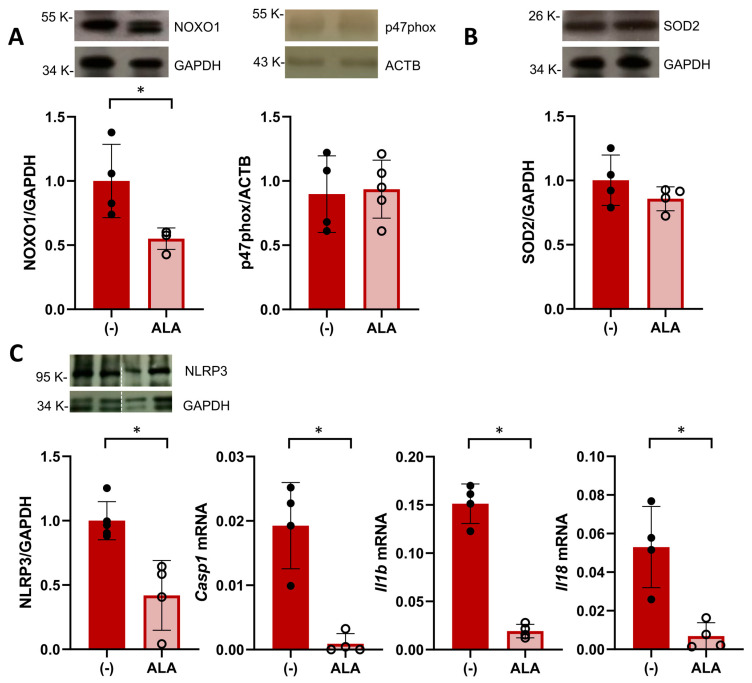
ALA attenuates placental oxidative stress and inflammasome priming. (**A**) Western blot analysis of placental expression of NOX subunits NOXO1 (left graph) and p47phox on GD14. Data are individual values and mean ± SD, *n* = 4–5 placentas/group, * *p* ˂ 0.05 analyzed by Mann–Whitney U-test. (**B**) Placental levels of SOD2 analyzed on GD14. Data are individual values and mean ± SD, *n* = 4–5 placentas/group (**C**) Placental expression of components of the NLRP3 inflammasome pathway in ALA-treated SHRSP. On GD20, treated rats displayed decreased protein levels of NLRP3 (analyzed by western blot, left graph) and mRNA expression of caspase 1 (Casp1), IL1β (Il1b), and IL18 (Il18). Data are individual values and mean ± SD, *n* = 4–6 placentas/group, * *p* ˂ 0.05 analyzed by Mann–Whitney U-test. Western blot and RT-PCR data are representative of 2 independent experiments run in duplicate.

**Figure 4 antioxidants-13-00730-f004:**
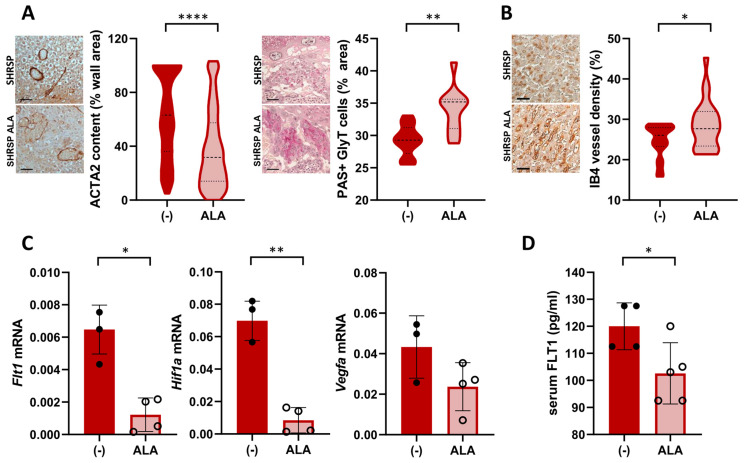
Treatment with ALA improves spiral artery remodeling, trophoblast differentiation, and angiogenic balance in SHRSP pregnancies. (**A**) Histological assessment of spiral artery remodeling (αSMA staining, left graph) and glycogen trophoblast (GlyT) content (PAS-positive stain, right graph) in GD14 placenta. Scale bar: 50 µm. ALA restored arterial remodeling (noted by loss of arterial wall expression of ACTA2) and GlyT differentiation in treated dams. Data are medians (dashed line) and minimum–maximum, *n* = 6 samples/group analyzing > 100 artery cross-sections or 10–15 fields at ×200 using QuPath. **** *p* ˂ 0.0001, ** *p* ˂ 0.01 analyzed by *t*-test (**B**) Analysis of placental labyrinth vascularization on GD20, based on isolectin B4 (IB4) staining. Scale bar: 50 µm. Vascular density (%IB4-positive stain) was increased upon ALA treatment. Data are medians (dashed line) and minimum–maximum, *n* = 8–9 samples/group analyzing 20–30 fields at ×200 using Angiotool. * *p* ˂ 0.05 analyzed by *t*-test. (**C**) Placental expression of Flt1 (Flt1, left graph), HIF1a (Hif1a, middle), and VEGF (Vegfa, right graph) analyzed by RT-PCR. Data are individual values and mean ± SD, *n* = 3–4 placentas/group, * *p* ˂ 0.05, ** *p* ˂ 0.01 analyzed by Mann–Whitney U-test. Results are representative of 2 independent experiments, run in duplicate. (**D**) Maternal serum FLT1 levels on GD20, determined by ELISA. Data are individual values and mean ± SD, *n* = 4–5 samples/group, * *p* ˂ 0.05 analyzed by Mann–Whitney U-test.

## Data Availability

The datasets used and/or analyzed during the current study are available from the corresponding author upon reasonable request.

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
