# Peer review of "Therapeutic Effect of Alpha Lipoic Acid in a Rat Preclinical Model of Preeclampsia: Focus on Maternal Signs, Fetal Growth and Placental Function"

_antioxidants, 2024, doi:10.3390/antiox13060730_

Round 1

Reviewer 1 Report (Previous Reviewer 1)

The problem discussed in the manuscript is interesting and important as well as for scientists and medicine practitioners but needs important changes.

1. The title - why did Authors used the term" Novel"

2. Why the experiment includes only   SHRSP rats? The comparison with "normal" rats is crucial.

3.The statement that the details of M&M are in the supplementary does not allow to review properly this manuscript.

4. The Conclusion is not valid-because the first sentence starts: In summary...

There are different paragraphs.

The figures are clear, as well as descriptions.

The References must be written in proper way.

Author Response

Comments for Authors

Major comments

The problem discussed in the manuscript is interesting and important as well as for scientists and medicine practitioners but needs important changes.

  1. The title - why did Authors used the term" Novel"

We thank the reviewer for this comment. Although we recognize that the therapeutic implications of ALA have been extensively discussed previously (as example, for recent reviews please refer to Salehi et al Biomolecules 2019, doi:10.3390/biom9080356 and Lv et al Int J Pharm 2022, doi: 10.1016/j.ijpharm.2022.122201) and have been tested in more than 150 clinical trials so far; our study is to our knowledge the first one to discuss its potential application as an intervention in hypertensive disorders of pregnancy. As our preclinical findings set the basis for future translational research that are pivotal to the repurposing of ALA as pharmacological treatment for pregnancy hypertension, we believe it is worth highlighting the novelty of this potential application in the title of our manuscript.  

  1. Why the experiment includes only SHRSP rats? The comparison with "normal" rats is crucial.

As already mentioned in the previous revision of the manuscript, our study included the blood pressure profile of the four experimental groups (WKY+PBS, WKY+ALA, SHRSP+PBS, SHRSP+ALA, presented on Table S3) for validation of the animal model used. Furthermore, we also evaluated the fetal growth parameters for the four groups (Results section, Figure S2) to show that ALA is able to restore fetal growth comparable to the normotensive animals. We believe this validation is important to demonstrate that the findings of our previous studies (Barrientos et al 2017, doi: 10.1093/molehr/gax024) are reproducible and that the SHRSP strain is a robust preclinical model for the pathophysiology of PE superimposed on chronic hypertension. However, since our manuscript is relevant to a possible therapeutic use of ALA in the hypertensive status, and also considering that the safety of oral alpha-lipoic acid treatment in normal pregnant women has already been established in a recent retrospective cohort (Parente et al Eur Rev Med Pharmacol Sci 2017 Sep;21(18):4219-4227), we believe the comparisons with control (WKY) pregnancies already included in our manuscript are sufficient to support our findings. Indeed, we want to point out that the comparison between SHRSP and WKY pregnancies has been extensively validated as a model of PE both by our group (doi: 10.1093/molehr/gax024) and also by others (Small et al 2015, doi: 10.1016/j.placenta.2015.10.022). Adding this data to the main text would imply making significant alterations to the manuscript, which has already been accepted in its present form by other two independent reviewers.

3.The statement that the details of M&M are in the supplementary does not allow to review properly this manuscript.

Again, as noted in the previous revision of our manuscript, referring the readers to previous publications or depositing extensive descriptions of methods in data supplements is a common, legitimate practice in scientific publishing. However, in an attempt to facilitate the assessment of our manuscript and improve readability, we have now expanded the M&M to provide more detail of the blood pressure determination method and fetal phenotype evaluation. These changes have been highlighted in red in the revised version of the manuscript. Only larger experimental protocols for qPCR and protein expression analyses are provided in the Supplemental Material.

  1. The Conclusion is not valid-because the first sentence starts: In summary...

There are different paragraphs.

We thank the reviewer for this comment. We have edited the Conclusions section as a single paragraph taking into account the reviewer’s suggestion.

Detail comments

The figures are clear, as well as descriptions.

The References must be written in proper way.

We thank the reviewer for the appraisal of the figures. As for the References, they have been formatted according to the Journal’s style requirements using Endnote.

Reviewer 2 Report (Previous Reviewer 2)

The authors addressed all my previous suggestions, and I have no further comments to add to my review. 

The authors addressed all my previous suggestions, and I have no further comments to add to my review. Thank you and congratulations!

Author Response

Thank you for the careful consideration given to our manuscript

Reviewer 3 Report (Previous Reviewer 3)

Thank you very much for implementing my suggestions. I think that the quality of the paper has improved and I recommend the publication.

no further comments

Author Response

Thank you for the careful assessment of our manuscript and positive feedback.

Reviewer 4 Report (New Reviewer)

This is a well-structured and commendable experimental study presented by a research group with expertise in this scientific field. The aims of the study and the study design are clear, realistic, and appropriate. The rat model for preeclampsia is well described and widely accepted. 

2.5. An explanation should be provided as to why placental gene expression analysis was conducted only at the mRNA level and why these results were not confirmed at the protein level. Other placental proteins were investigated by Western blot (2.4) This should be clarified.

Author Response

Thank you for your appraisal of our manuscript

Thank you for your appraisal of our manuscript. We have analysed oxidative stress marker proteins (i.e., NADPH oxidase subunits) by western blot because we had already established the protocols for our previous publication (Blois et al Cellls 2021, doi: 10.3390/cells10040800). For this paper, we added expression analyses of inflammasome components (western blot for NLRP3 and mRNA for the rest of the markers, we also analysed IL18 protein in the amniotic fluid) and angiogenic regulators (i.e., Flt-1, which in addition to mRNA levels was determined by ELISA in the maternal circulation). As activation of the NLRP3 inflammasome involves both transcriptional regulation (priming stage) and protein translation and processing (activation), we believe our approach is valid to investigate the effect of lipoic acid on placental oxidative stress, inflammatory and angiogenic responses.

Round 2

Reviewer 1 Report (Previous Reviewer 1)

Dear Authors,

Your responses to my questions are not satisfied. I was asking for reviewing THIS manuscript not ALL of your previous publications, so I need to have all of the information in THIS manuscript.

The same as in Review 1

Author Response

Dear Reviewer 1,

Thank you for your appraisal of our manuscript. We are sorry that you were not satisfied by our response. Since the Editor and three other independent reviewers provided positive feedback on our work, we have no further comments.

This manuscript is a resubmission of an earlier submission. The following is a list of the peer review reports and author responses from that submission.

Round 1

Reviewer 1 Report

Major comments:

1. The study does not have control group - without hypertension

2. Autoplagiarism

Due to two serious concerns I will not specify detail comments.

Reviewer 2 Report

Overall, the manuscript titled "Novel therapeutic effect of alpha lipoic acid in a rat preclinical model of preeclampsia: focus on maternal signs, fetal growth and placental function" presents a comprehensive investigation into the potential therapeutic role of alpha lipoic acid (ALA) in a rat model of preeclampsia (PE). Below are some detailed comments and suggestions for improvement:

-  The abstract transitions smoothly between introducing the study's background, presenting the hypothesis, outlining the methodology, and summarizing the key findings. However, you could consider adding a sentence or phrase to bridge the gap between describing the methodology and summarizing the results. Also, it could be improved by including specific quantitative results to give readers a clearer understanding of the findings.

-       In the introduction, it might be beneficial to include a brief overview of previous studies investigating antioxidant therapies in preeclampsia to provide context for the rationale behind studying ALA.

-    The methods section is detailed and includes information on animal models, experimental design, and analytical techniques, being essential for reproducibility. It would be helpful to include information on sample size calculations or statistical power analyses to justify the chosen sample sizes and ensure adequate power to detect significant effects.

-    The results section is comprehensive and presents data on maternal blood pressure, renal function, fetal growth, placental morphology, and molecular analyses. Figures and tables effectively complement the text. Quantitative data are provided, which is essential for understanding the magnitude of the observed effects. However, some data could be presented more clearly, perhaps through the use of additional summary statistics or graphical representations.

-     The discussion provides a thorough interpretation of the results in the context of existing literature. It effectively highlights the significance of the findings and potential mechanisms underlying the therapeutic effects of ALA. It might be beneficial to discuss any limitations of the study, such as potential confounding factors or limitations inherent to the animal model used. Addressing these limitations could help readers interpret the findings more accurately.

-     The conclusions succinctly summarize the key findings of the study and suggest future research directions. They effectively highlight the potential clinical relevance of the findings. It would be useful to reiterate the implications of the study for clinical practice and the potential impact on patient care.

Reviewer 3 Report

Regarding statistical analysis and study design: Some of them are not appropriate because of the rather small smaple sizes. Also the study design raises questions.

Regarding statistical analysis and study design:

1. How have the rats been assigned to one of the two groups? Randomly?

2. How has the sample size been estimated?

3. Results regarding quantitative variables are expresses as mean +/- standard error SEM. SEM however depends from the sample size. Instead of SEM, standard deviation should be given.

4. Shapiro-Wilk test has been used to tst for normal distribution. However, because of the rather small sample sizes (i.e. n=5) this test has not enpugh power to detect that the sample data do not fit to a normal distribution. Because of the rather small samples sizes, Mann Whtney U test test should be preferred.

5. ROUT method was used to identify outliers. As thiis method is not generally known, it should be briefly explained.

6. Furthermore: How many outliers have been detected? What was the reason for that? What happend to the outliers? It is not ok to remove outliers from the data analysis without a valid reason.

7. The legends of the figures are somewhat confusing. In my opinion it would be better to present the details (i.e. mean values, p values) in tables.

8. In some places, only the information "no differences" is given. Also for results which are not statistically significant, the corresponding p value should be given.

9. Line 136: The term "parametric and non-parametric data" is not correct. The tests are parameteric / not parametric. The 2 sample t test is a parametric test which may be used for dataapproximately normally distributed. For skewd data, Mann-Whitney U test as a non-parametric test (ranksum test) should be preferred.